# Attention Deficit Hyperactivity and Autism Spectrum Disorders as the Core Symptoms of AUTS2 Syndrome: Description of Five New Patients and Update of the Frequency of Manifestations and Genotype-Phenotype Correlation

**DOI:** 10.3390/genes12091360

**Published:** 2021-08-30

**Authors:** Carolina Sanchez-Jimeno, Fiona Blanco-Kelly, Fermina López-Grondona, Rebeca Losada-Del Pozo, Beatriz Moreno, María Rodrigo-Moreno, Elena Martinez-Cayuelas, Rosa Riveiro-Alvarez, María Fenollar-Cortés, Carmen Ayuso, Marta Rodríguez de Alba, Isabel Lorda-Sanchez, Berta Almoguera

**Affiliations:** 1Department of Genetics and Genomics, IIS–Fundación Jiménez Díaz University Hospital, 28040 Madrid, Spain; carolina.sanchez@fjd.es (C.S.-J.); fblancok@quironsalud.es (F.B.-K.); fermina.lopez@quironsalud.es (F.L.-G.); RRiveiro@fjd.es (R.R.-A.); CAyuso@fjd.es (C.A.); mrodrigueza@fjd.es (M.R.d.A.); ilorda@fjd.es (I.L.-S.); 2Center for Biomedical Network Research on Rare Diseases (CIBERER), ISCIII, 28040 Madrid, Spain; 3Department of Pediatrics, IIS–Fundación Jiménez Díaz University Hospital, 28040 Madrid, Spain; rebeca.losada@quironsalud.es (R.L.-D.P.); beatriz.moreno@fjd.es (B.M.); MRodrigoMo@fjd.es (M.R.-M.); elena.martinezc@quironsalud.es (E.M.-C.); 4Clinical Genetics Unit, Department of Clinical Analysis, Clínico San Carlos University Hospital, 28040 Madrid, Spain; mariadelmar.fenollar@salud.madrid.org; 5IIS-Clínico San Carlos University Hospital (IsISSC), 28040 Madrid, Spain

**Keywords:** *AUTS2*, AUTS2 syndrome, ADHD, neurodevelopmental disorder, autism

## Abstract

Haploinsufficiency of *AUTS2* has been associated with a syndromic form of neurodevelopmental delay characterized by intellectual disability, autistic features, and microcephaly, also known as AUTS2 syndrome. While the phenotype associated with large deletions and duplications of *AUTS2* is well established, clinical features of patients harboring *AUTS2* sequence variants have not been extensively described. In this study, we describe the phenotype of five new patients with *AUTS2* pathogenic variants, three of them harboring loss-of-function sequence variants. The phenotype of the patients was characterized by attention deficit/hyperactivity disorder (ADHD) and autism spectrum disorder (ASD) or autistic features and mild global developmental delay (GDD) or intellectual disability (ID), all in 4/5 patients (80%), a frequency higher than previously reported for ADHD and autistic features. Microcephaly and short stature were found in 60% of the patients; and feeding difficulties, generalized hypotonia, and ptosis, were each found in 40%. We also provide the aggregated frequency of the 32 items included in the AUTS2 syndrome severity score (ASSS) in patients currently reported in the literature. The main characteristics of the syndrome are GDD/ID in 98% of patients, microcephaly in 65%, feeding difficulties in 62%, ADHD or hyperactivity in 54%, and autistic traits in 52%. Finally, using the location of 31 variants from the literature together with variants from the five patients, we found significantly higher ASSS values in patients with pathogenic variants affecting the 3′ end of the gene, confirming the genotype-phenotype correlation initially described.

## 1. Introduction

Neurodevelopmental disorders are highly complex disorders characterized both by clinical and genetic heterogeneity and with a complex genetic architecture [1]. In addition, most genes described to date only affect a small subset of patients, often displaying variable expressivity between individuals, even within the same family [2,3]. Therefore, the specific phenotype associated with such genes is usually difficult to ascertain, and the interpretation and classification of genetic variants involved challenging.

*AUTS2* is an example of the above-mentioned complexity. Disruption of *AUTS2* causes a syndromic form of intellectual disability (ID) known as AUTS2 syndrome (OMIM #615834), characterized by a highly variable phenotype consisting of global developmental delay (GDD) and/or ID commonly associated with the combination of microcephaly, short stature, feeding difficulties, and hypotonia, as well as recognizable facial dysmorphic features [4,5,6].

The large interindividual, and even intrafamilial, variability observed in patients with *AUTS2* pathogenic variants led Beunders and colleagues in 2013 to establish what they called an AUTS2 syndrome severity score (ASSS), and that has been systematically used in the literature to assess the severity and phenotype of patients with AUTS2 syndrome [6]. The ASSS is based on 32 features found with a frequency of over 10% in the first cohort of patients described with *AUTS2* aberrations and includes items belonging to growth and feeding, neurodevelopment, neurologic disorders, dysmorphic features, skeletal abnormalities, and congenital anomalies [6].

A genotype-phenotype correlation between the severity of the syndrome by means of ASSS and the location of the *AUTS2* genetic alteration has been established [6]. The 19 exons of *AUTS2* are divided into a non-conserved 5′ region (N-terminal) that includes the first 8 exons and the more conserved 3′ end of the gene, which includes exons 9 to 19. There is a shorter isoform resultant from an alternative transcription start site in exon 9, consisting of the last 11 exons of the 3′ end and which is expressed in the human brain [6,7]. Disruption of the 3′ end of the gene, which also disrupts the short isoform, and disruption of the entire gene has been associated with a more severe phenotype and a complete AUTS2 syndrome, whereas a milder clinical phenotype and lower ASSS have been associated with the disruption at the 5′ end [5,6]. However, this correlation was described in the first cohort of patients [6], and as new patients with AUTS2 syndrome have been reported, this association has become less evident [4,8,9].

Since the identification of the gene and the description of the phenotypic characteristics of AUTS2 syndrome [6,10], more than 60 patients with pathogenic variants have been reported in the literature [4,5,6,8,9,10,11,12,13,14,15,16,17,18,19,20,21,22,23]. Most variants reported to date are de novo intragenic deletions [4,5,6], involving one or more exons, while pathogenic sequence variants in *AUTS2* only represent a small fraction of the *AUTS2* mutational spectrum. To date, only 13 sequence variants (nonsense single nucleotide variants, frameshift, and in-frame insertions and deletions) have been reported in the scientific literature [4,5,9,11,19,20,21,22,23,24], and 4 intragenic or exonic duplications [17,25], and thus the phenotype associated is less known. The ClinVar database currently has 96 variants in *AUTS2* classified as pathogenic or likely pathogenic (accessed on 15 July 2021), of which 34 are sequence variants (missense, nonsense, inframe and frameshift deletions and duplications and/or acceptor/donor splice site variants) and 62 are deletions (*N* = 53) or duplications (*N* = 9). However, ClinVar exclusively reports the interpretation and classification of the genetic variants, and no description of the clinical phenotype is provided.

In this study, we describe the phenotype of five new patients with *AUTS2* pathogenic variants, three of them harboring loss-of-function sequence variants, characterized by developmental delay, autistic features, and attention deficit hyperactivity disorder (ADHD). We also provide the aggregated frequency of the ASSS list of items in the cohorts previously published and confirm the genotype-phenotype correlation initially proposed by Beunders and colleagues [6], calculated with ASSS from our patients and patients reported in the literature.

## 2. Materials and Methods

### 2.1. Selection of Patients

The project was approved by the ethics committee of Fundacion Jimenez Diaz Hospital and was performed in accordance with the Declaration of Helsinki Principles and institutional requirements. Written informed consent was obtained from each participant or their guardians.

The selection of subjects was performed by retrospective review of the cohort of over 2000 patients with neurodevelopmental disorders from Fundacion Jimenez Diaz Hospital. Patients with a diagnosis of any form of neurodevelopmental disorder (GDD, ID, autistic features, or ADHD) with a pathogenic or likely pathogenic variant in *AUTS2* were selected. Patient information was extracted from the patients’ electronic health records. Clinical diagnosis of the patients was performed by a pediatric neurologist and patients were evaluated by a clinical geneticist, which included a detailed anamnesis, pedigree analysis, and physical examination.

### 2.2. Genomic Tests

Genomic DNA from all patients was extracted from peripheral blood samples using automated DNA extractors: BioRobot EZ1 (QIAGEN, Hilden, Germany). Parental samples were obtained to determine the origin of the genetic variants identified in the probands.

#### 2.2.1. aCGH

aCGH was performed using the aCGX 60K platform (CGXTM, PerkinElmer, Inc, Waltham, MA, USA) following the manufacturer’s protocol. Quality control included DLR spread, reproducibility, background/signal intensity, and signal/noise ratio. The array images were scanned and extracted using the SureScan Microarray Scanner (Agilent Technologies, Santa Clara, CA, USA). CNV analysis was conducted with the Genoglyphix^®^ platform (PerkinElmer, Inc, Waltham, MA, USA).

#### 2.2.2. Clinical Exome Sequencing and Variant Analysis

We used the Clinical Exome Solution v2 by Sophia Genetics (CES; Sophia Genetics, Boston, MA, USA) that targets 4490 genes involved in human diseases and the libraries were run on a NextSeq500 instrument (Illumina, San Diego, CA, USA). Next Generation Sequencing (NGS) data analysis was performed using algorithms developed by Sophia Genetics and implemented in the SOPHiA DDM™ analysis platform. Quality control of the NGS data and variant analysis was performed as previously described [26], and variant interpretation and classification followed the ACMG guidelines. *AUTS2* genetic variants were referred to as transcript NM_015570.

#### 2.2.3. Sanger Sequencing

Sanger sequencing was performed for family segregation of *AUTS2* variants identified by clinical exome sequencing.

### 2.3. Data Analysis

ASSS of the five patients were calculated as previously described [6]. Data on ASSS values from patients from the literature were extracted to perform a genotype-phenotype correlation between the scores and the location of the variant on the 5′ or 3′ region of the gene. For that, the median ASSS values and the standard deviation (SD) from all patients from our cohort and the literature, grouped by the location of the *AUTS2* variants, were calculated. A Mann–Whitney test was used for the association. Significance was set as *p* < 0.05.

A literature review of AUTS2 syndrome patients was also performed to calculate the frequency of the features described in the syndrome. For that purpose, only papers with a comprehensive description of the phenotype were considered. Frequencies were calculated as the percentage of patients with a specific item from the total of patients evaluated for that item.

## 3. Results

### 3.1. Clinical and Molecular Characteristics of the Five Patients with AUTS2 Pathogenic Variants: ADHD, Autistic Traits, and GDD/ID as the Main Features

We identified five patients with pathogenic or likely pathogenic de novo variants in *AUTS2* by aCGH (in 2 patients) and by clinical exome sequencing (in 3 patients). A total of 80% of the patients were males (4 of 5), and the mean age (±SD) at diagnosis was 7 (±4.3) years, ranging from 14 months to 12 years. Variants identified and phenotypes of the five patients, along with the ASSS values, are displayed in Table 1.

Four of the five variants identified were novel and one has been previously reported in ClinVar. Regarding the location on the gene, four variants were located in the N-terminal region: arr[hg19] 7q11.22 (67767963_69320956) × 3 in exon 1 (RM-1003), arr[hg19] 7q11.22 (69564262-69592731) × 1in exon 3 (RM-299); c.927_928delinsAT; p.Gln310* in exon 7 (RM-1935); and c.1298del; p.Leu433Profs*40 in exon 8 (RM-1513); while one was in the C-terminal end:c.2183del; p.Gly728Alafs*2 in exon 17 (RM-519) (Table 1).

Indication for genetic testing of the patients was primarily ADHD or inattention, accompanied by other neurodevelopmental or behavioral symptoms in 4 of the 5 patients. The fifth patient (RM-1935) was referred to the genetics department at 14 months of age due to presenting GDD, global hypotonia, and failure to thrive (Table 1).

The main clinical features of the patients were ADHD and ASD or autistic features and mild GDD or ID, all in 4/5 patients (80%), and two patients had a history of language and motor delay (RM-1003 and RM-519) (Table 1). Microcephaly and short stature were found in three patients (60%); and feeding difficulties, generalized hypotonia, and ptosis were each found in two of the patients (40%). Three of the five patients had two café-au-lait spots (RM-299, RM-519, RM-1003).

ASSS values were calculated for the five patients, and values are displayed in Table 1.

### 3.2. Frequency of AUTS2 Syndrome Features: Literature Review

We calculated the frequency of the 32 items included in the ASSS in patients currently reported in the literature. For that, only 9 studies with a comprehensive description of the phenotype were considered: papers by Beunders et al., 2016 [5], which includes aggregated data from their two previous reports [4,6] as well as from other authors [10,14,15,16,17,27,28,29] and other seven studies [8,9,11,18,19,20,21]. The total number of patients reported by such studies was 61 but the number of patients evaluated for the 32 items was highly variable and ranged from 11 to 61 (Table 1). The main characteristics of the syndrome are GDD/ID in 98% of patients, microcephaly in 65%, feeding difficulties in 62%, ADHD or hyperactivity in 54%, and autistic traits in 52%. The rest of the features had a frequency of less than 50% (Table 1).

### 3.3. Genotype-Phenotype Correlation between ASSS Values and Location of the Variant Confirmed on 36 Patients

A review of the literature retrieved seven studies reporting on 31 patients with AUTS2 syndrome with ASSS values and an *AUTS2* pathogenic variant (Table 2; [4,6,8,9,16,20,21]). The 31 variants from the literature together with variants from the five patients, were grouped based on whether the location was N-terminal (exons 1 to 8; *N* = 26) or C-terminal (exons 9 to 19; *N* = 10). Median ASSS values (±SD) of patients with variants in the N-terminal region were 8.5 (±5.2) and 15 (±4.8) for variants in the C-terminal region. Comparison of the median values of both groups using a Mann–Whitney test was statistically significant (*p* = 0.03).

## 4. Discussion

More than 60 patients with AUTS2 syndrome have been described to date [4,5,6,8,9,10,11,12,13,14,15,16,17,18,19,20,21,22,23]. These patients mostly carry de novo intragenic or exonic deletions, whereas loss-of-function small variants are not as frequently found in the literature [4,5,9,11,19,20,21,22,23,24]. In this study, we add to the existing knowledge of the syndrome with the description of the phenotype of five new patients with *AUTS2* pathogenic variants, of which three are loss-of-function small variants.

In our cohort, we found that, except for one patient who was too young to be adequately assessed (RM-1935), all our patients had a diagnosis of ADHD and or hyperactivity/inattention and displayed autistic features, and that 80% had a global developmental delay or mild intellectual disability. Other features frequently seen were microcephaly and short stature, both in 60% of the patients, as well as café-au-lait spots, not described before. Feeding difficulties, generalized hypotonia, and ptosis were found in 40% of the patients. To assess whether these numbers were consistent with those previously reported, we calculated the frequency of the 32 items included in the ASSS on 61 patients from the literature [5,10,14,15,16,17,27,28,29]. Percentages seen in our cohort are consistent with the global frequency found in the literature, although features such as ADHD and autistic traits were more commonly found in our five patients (100% versus 54% for ADHD and 52% for ASD/autistic features), being both conditions the clinical indications for the genetic study in four of the five patients.

Notably, not all our patients had GDD or ID: RM-299, who carried a duplication of exon 3 of *AUTS2*, only displayed ADHD and behavioral symptoms. To date, there are only four patients reported in HGMD with a pathogenic intragenic duplication of *AUTS2*, three involving exon 5, and all with developmental delay or ASD [17,25,30]. Ben-David et al. reported a duplication in exon 5 of *AUTS2* that resulted in the monoallelic expression of the gene, supporting a pathogenic effect of intragenic duplications [25].

The rest of the features initially described by Beunders in 2013 [6], and which constitute the ASSS, were found in our cohort in a frequency of 25% or less, being the majority not found in our sample (16 of the 32 items). This results in low ASSS values in our cohort, ranging from 2 to 11. The highest ASSS, 11, was found in patient RM-519, who carried a loss-of-function variant in exon 17, located in the C-terminal region of the gene. To the best of our knowledge, this is the first report of a pathogenic variant in exon 17, which, based on the initial observations by Beunders et al. in 2013, is expected to result in a more severe phenotype [6]. This patient was the only one that was comprehensively examined, with the list of items from the ASSS, after obtaining the genetic results and his score did not change substantially: his phenotype was mild and limited to growth and neurodevelopment.

Even though the number and phenotype of patients with AUTS2 syndrome has increased since the first report, the genotype-phenotype correlation has not been updated since then, despite the overlap of ASSS values from patients with alterations of the 5′ and 3′ ends [4,6,8,9,16,20,21]. In order to confirm this genotype-phenotype correlation, we collected data on patients with ASSS values published in the literature and, along with our five patients, tested the association between the location of the variant (exons 1 to 8 versus 9 to 19) and such values. We found significantly higher ASSS values in patients with pathogenic variants affecting the 3′ end of the gene, confirming the genotype-phenotype correlation initially described [6]. However, the range of ASSS scores in patients with AUTS2 syndrome is quite broad (0–17 for the N-terminal and 11–22 for the C-terminal), and values overlap in the two groups.

It must be noted that half of the items from the ASSS relate to the dysmorphic features (16 of 32) initially reported by Beunders and colleagues in their first series of patients and that they suggested of being characteristic of the syndrome [4,5,6]. However, this facial phenotype has not been consistently described in patients with AUTS2 syndrome, and neither was found in our patients. Identification of dysmorphic facial features, which are sometimes very subtle, is highly dependent on the experience of the observer and the age of the patients, and which in younger patients may not be evident. Age is also a factor affecting the assessment of neurodevelopmental conditions such as ASD, ID, or ADHD. One example of this is patient RM-1935, who was 14 months at the time of testing, an age when ASD and ADHD cannot be properly diagnosed clinically. Indeed, this patient was the only one of the five who did not display either ASD or ADHD in the entire cohort. Another factor affecting ASSS assessment is that often patients are reevaluated retrospectively upon genetic diagnosis, and thus ASSS values are also dependent upon the availability of phenotypic data in the clinical history.

The list of items currently included in the ASSS may not accurately reflect the severity of the syndrome but rather whether the phenotype is more or less complete and dysmorphic. In this regard, Gieldon and coworkers recently highlighted this limitation of the ASSS and proposed to adapt the score to contemplate the severity of the main features such as ID or ASD [9]. We agree with this observation, which is further supported by the description of patients with similar ASSS and large differences in the severity of the phenotype [6,21]. An example of such differences is patient S6 described in the series reported by Beunders and colleagues in 2013 and the patient described by Martinez-Delgado and coworkers [6,21].These patients have ASSS values of 17 and 15, respectively, and while values from the former patient came, mostly, from dysmorphic and general features (short stature, feeding difficulties, and microcephaly) and had mild ID, the second was severely affected from birth with significant craniofacial dysmorphism, as well as severe growth and neurodevelopmental delay, with severe language disorder, and other neurological manifestations such as global hypotonia and structural brain defects [21]. This example supports the need to account for the severity of manifestations, especially the most disabling conditions such as ID, ASD, or neurological disorders in the ASSS. Severity could be accounted for by adding extra points based on whether items included in the categories of “neurodevelopment” and “neurological disorders” from the ASSS are mild (i.e., 1 point), moderate (i.e., 2 points), or severe (i.e., 3 points). On the other hand, the rating of items from the categories that relate to the phenotype of patients (“skeletal and limb abnormalities” and “dysmorphic features”) could be adjusted to lower punctuation or considered in a separate scale.

As previously mentioned, ASSS scores in our cohort were low and the phenotype of the patients was primarily restricted to growth and neurodevelopment, with all patients presenting mild phenotypes. This milder form of AUTS2 syndrome may be the consequence of patients carrying pathogenic sequence variants, while intragenic deletions could lead to a more complete form of the syndrome, regardless of the location of the alteration. However, this hypothesis cannot be currently tested due to the low number of patients reported in the literature with pathogenic sequence variants in *AUTS2*. This limited number compared to that with large deletions might be due to the previous extended use of aCGH in the diagnosis of neurodevelopmental disorders [31,32]. Nowadays, due to the higher diagnostic yield, NGS is considered the diagnostic method of choice for ID/GDD [26,33], and as whole exome or genome sequencing becomes more accessible to the genetics laboratories, more patients with pathogenic variants at the sequence level will be identified. Delineation of the phenotype associated with these variants is critical in the assessment and genetic counseling of these patients.

## 5. Conclusions

In conclusion, we add to the current knowledge of AUTS2 syndrome with the description of the phenotype of five new patients, of whom three carry pathogenic sequence variants in the gene, provide updated frequencies of the clinical characteristics of the syndrome based on 61 patients previously published, and confirm the genotype-phenotype correlation initially reported.

## Figures and Tables

**Table 1 genes-12-01360-t001:** Molecular and Clinical Characteristics of the Patients Included in the Study, Based on the ASSS List of Items. The frequency of the AUTS2 syndrome features in our cohort and the literature is also displayed. Data for the latter was extracted and calculated from (references). The ASSS from our patients is included at the end of the table. For the age at diagnosis, y = years and m = months. GDD = global developmental delay; ID = intellectual disability; ASD = autism spectrum disorders; ADHD = attention deficit/hyperactivity disorder; NA = not assessed.

	RM-1003	RM-299	RM-1935	RM-1513	RM-519		
Variant	arr[hg19] 7q11.22 (67767963_69320956) × 3	arr[hg19] 7q11.22 (69564262-69592731) × 1	c.927_928delinsAT; p.Gln310*	c.1298del; p.Leu433Profs*40	c.2183del; p.Gly728Alafs*2		
Exon	1	3	7	8	17		
Novel/Described	Novel (reported in [26])	Novel (reported in [26])	Novel	ClinVar and reported in [26]	Novel		
De novo	1	1	1	1	1		
Age at diagnosis	5 y 5 m	12 y 1 m	16 m	7 y 6 m	10 y 7 m		
Indication for genetic study	Developmental delay and inattention	Behavioral problems, ADHD	GDD, hypotonia, failure to thrive	Microcephaly, ADHD	Cognitive delay, ADHD symptoms. motor stereotypies	Frequency in our cohort (%)	Frecuency in the literature (%)
General							
Low birth weight	-	-	-	-	-	0%	20.4% (10 of 49)
Short stature	1	-	1	-	1	60%	42.6% (23 of 54)
Microcephaly	1	-	-	1	1	60%	65.4% (34 of 52)
Feeding difficulties	-	-	1	-	1	40%	62.0% (31 of 50)
Neurodevelopmental							
GDD/ID	1	-	1	1	1	80%	98.4% (60 of 61)
ASD/autistic features	1	1	NA	1	1	100%	51.9% (12 of 36)
Sound sensitivity	-	-	NA	-	1	20%	33.3% (27 of 52)
Hyperactivity/ADHD	1	1	NA	1	1	100%	54.2% (13 of 24)
Neurological disorders							
Generalized hypotonia	-	-	1	-	1	40%	38.2% (21 of 55)
Structural brain anomaly	-	-	-	-	-	0%	26.8% (11 of 41)
Cerebral palsy, spasticity, high muscle tone	-	-	1	-	-	20%	36.5% (19 of 52)
Skeletal and limb anomalies							
Kyphosis/scoliosis	-	-	-	-	-	0%	23.8% (10 of 42)
Arthrogryposis/shallow palmar creases	-	-	-	-	-	0%	26.1% (6 of 23)
Tight heel cords	-	-	-	1	-	20%	9.1% (1 of 11)
Congenital malformations							
Hernia umbilicalis	-	-	-	-	-	0%	11.1% (6 of 54)
Patent foramen ovale/ASD	-	-	1 (PFO)	-	-	20%	4.2% (1 of 24)
Dysmorphic Features							
Highly arched eyebrows	-	-	-	-	1	20%	37.5% (12 of 32)
Hypertelorism	-	-	-	-	-	0%	43.8% (14 of 32)
Proptosis	-	-	-	-	-	0%	21.9% (7 of 32)
Short palpebral fissures	-	-	1	-	-	20%	25.0% (8 of 32)
Upslanting palpebral fissures	-	-	-	-	-	0%	15.6% (5 of 32)
Ptosis	-	-	-	1	1	40%	28.1% (9 of 32)
Epicanthal fold	-	-	-	-	-	0%	25.0% (8 of 32)
Strabismus	-	-	1	-	-	20%	25.0% (8 of 32)
Prominent nasal tip	-	-	-	-	1	20%	18.8% (6 of 32)
Anteverted nares	-	-	-	-	-	0%	21.9% (7 of 32)
Deep and/or broad nasal bridge	-	-	-	-	-	0%	37.5% (12 of 32)
Short and/or upturned philtrum	-	-	-	-	-	0%	34.4% (11 of 32)
Micrognathia/retrognatia	-	-	-	-	-	0%	35.5% (11 of 31)
Low-set ears	-	-	-	1	-	20%	32.3% (10 of 31)
Earpit	-	-	-	-	-	0%	16.1% (5 of 31)
Narrow mouth	-	-	-	-	-	0%	50.0% (16 of 32)
ASSS	5	2	8	6	11		
Other features							
	2 CAL spots	Oppositional defiant disorder, aggressiveness, tics. 2 CAL spots	Narrow and downslanting palpebral fissures, short nose, dentition delay, hypermetropy, dysphagia, and sleep disorder	Dolicocephaly, peculiar helix, prognathism, and clubfoot	2 CAL spots	--	

**Table 2 genes-12-01360-t002:** ASSS Scores from AUTS2 Syndrome Patients from the Literature and our Cohort Sorted by the Location of the Variant on the N-terminal and C-terminal ends of the Gene.

Reference	Exon Number	ASSS N-Terminal (1–8)	ASSS C-Terminal (9–19)
[6] P1	2	1	
[6] F1	2	0	
[6] P2	3–4	5	
[6] P3	3–4	6	
[6] P4	3–4	6	
[6] M4	3–4	5	
[6] P5	1–4	11	
[6] P6	5–6	16	
[6] S6	5–6	17	
[6] M6	5–6	8	
[6] P7	6	9	
[6] P8	6	8	
[6] P9	6–9		22
[6] P10	6–11		15
[6] P11	6–18		8
[6] P12	7–19		9
[6] P13	7–19		18
[6] P14	All		7
[6] P15	All		16
[6] P16	4	11	
[6] P17	6	17	
[4] P1	7	12	
[4] P2	6	16	
[8] P1	6	17	
[8] P2	12–19		15
[8] P3	6	17	
[16]	1	3	
[20]	8	15	
[9] P1	6	7	
[9] P2	6	13	
[21]	9		15
Patients from the present study			
RM-1003	1	5	
RM-299	3	2	
RM-1935	7	8	
RM-1513	8	6	
RM-519	17		11
Median (± SD)		8.5 (± 5.2)	15 (± 4.8)

## Data Availability

N/A.

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
