# Peer review of "Attention Deficit Hyperactivity and Autism Spectrum Disorders as the Core Symptoms of AUTS2 Syndrome: Description of Five New Patients and Update of the Frequency of Manifestations and Genotype-Phenotype Correlation"

_genes, 2021, doi:10.3390/genes12091360_

Round 1

Reviewer 1 Report

In this paper the authors describe the phenotype of five new patients with AUTS2 pathogenic variants. They also provide the aggregated frequency of the AUTS2 syndrome severity score (ASSS) list of items in the cohorts previously published and confirm the genotype-phenotype correlation initially proposed by Beunders and colleagues, calculated with ASSS from their patients and patients reported in the literature.

The manuscript is well written and of interest. Although only five new patients are reported, this study broadens the current knowledge of AUTS2 syndrome.

Author Response

We thank the reviewer for the comments.

Reviewer 2 Report

Overall, this is a well written and structured manuscript about a very interesting topic.

Author Response

We thank the reviewer for the comments. In the table, this reviewer indicates that the presentation of results could be improved. However, no detail about why or how to improve presentation is provided

Reviewer 3 Report

This research article describes the phenotypes of five patients with pathogenic sequence variants in AUTS2, variation in which has been associated with the AUTS2 syndrome involving intellectual disability, symptoms of autism spectrum disorder, and microcephaly. The authors also assess the frequency individual items from the AUTS2 syndrome severity score (ASSS) from case reports in the current literature. Finally, they examine the location of 31 variants from the literature and the variants identified in the five patients described here. Consistent with prior literature, the authors found higher ASSS values in patients with variants affecting the 3’ end of the gene.

The article is well-written, and covers an important topic considering that there are few published data about the phenotype associated with AUTS2 sequence variants. The methods described are reasonable and the authors are careful not to overstate their claims. Though the small sample size (n=5) limits the conclusions that can be drawn, overall, the article should make a nice addition to the literature. I have a few minor comments and suggestions, which are listed below:

Minor Comments:

  1. There seems to be a copy-paste error in lines 138-145 – these two paragraphs should be revised.
  2. Line 151: Typo
  3. The authors appropriately note issues with the ASSS criteria, including the challenge of applying them to retrospective phenotype reports and the questionable correlation of some of the items with syndrome severity. I would be interested to hear the authors’ thoughts on how the ASSS could be adapted in light of their findings.

Author Response

We thank the reviewer for the positive comments about our manuscript and the suggestions. Below is a point-by-point response to the comments:

1. There seems to be a copy-paste error in lines 138-145 – these two paragraphs should be revised.

The paragraphs do not belong to that section and have been removed

2. Line 151: Typo

We are unsure where the typo is

3. The authors appropriately note issues with the ASSS criteria, including the challenge of applying them to retrospective phenotype reports and the questionable correlation of some of the items with syndrome severity. I would be interested to hear the authors’ thoughts on how the ASSS could be adapted in light of their findings.

A paragraph with our suggestions about how the ASSS could be modified has been included:

"This example supports the need to account for the severity of manifestations, especially the most disabling conditions such as ID, ASD or neurological disorders, in the ASSS. Severity could be accounted for by adding extra points based on whether items included in the categories of “neurodevelopment” and “neurological disorders” from the ASSS are mild (i.e. 1 point), moderate (i.e. 2 points), or severe (i.e. 3 points). On the other hand, rating of items from the categories that relate to the phenotype of patients (“skeletal and limb abnormalities” and “dysmorphic features”) could be adjusted to a lower punctuation or considered in a separate scale"